# Effect of Dietary Selenium on the Growth and Immune Systems of Fish

**DOI:** 10.3390/ani13182978

**Published:** 2023-09-20

**Authors:** Sahr Lamin Sumana, Huangen Chen, Yan Shui, Chengfeng Zhang, Fan Yu, Jian Zhu, Shengyan Su

**Affiliations:** 1Wuxi Fisheries College, Nanjing Agricultural University, Wuxi 214081, China; sl5284sumana@gmail.com (S.L.S.); shuiyan@ffrc.cn (Y.S.); zhuj@ffrc.cn (J.Z.); 2Jiangsu Fishery Technology Promotion Center, Nanjing 210017, China; 13705151811@163.com; 3Key Laboratory of Integrated Rice-Fish Farming Ecology, Ministry of Agriculture and Rural Affairs, Freshwater Fisheries Research Center, Chinese Academy of Fishery Sciences, Wuxi 214081, China; zhangchengfeng@ffrc.cn (C.Z.); yufan@ffrc.cn (F.Y.)

**Keywords:** dietary Se effects, growth performance, immune functions, fish species, aquatic environment

## Abstract

**Simple Summary:**

This paper addresses the various sources of Se in the aquatic environment as well as the positive and negative effects on fish. It emphasizes that optimal dietary Se levels are necessary for healthy biological processes in fish, such as growth, reproduction, and immunity. Since organic Se appears to be the most ideal for fish due to its low toxicity, environmental safety, and efficient fish culture, it explores the potential sources and forms of Se. Extreme doses could be toxic to fish, and higher levels could cause retarded growth, survival issues, abnormalities, and poor performance. However, adverse effects depend on the acceptance of the fish species. Additionally, there are a number of biological techniques that may be used to remove excess Se from the aquatic environment, with phytoremediation being the most effective. The production of Se-rich feeds and their benefits for fish immune systems, disease prevention, and growth development are discussed in this article. These factors contribute to the profitability of fish farmers and their confidence in the feed industry. In order to monitor and study Se’s effects on fish and their aquatic environment, the report highlights the significance of the feed industry and how it connects farmers with research institutions.

**Abstract:**

Dietary selenium (Se) is an essential component that supports fish growth and the immune system. This review attempts to provide insight into the biological impacts of dietary Se, including immunological responses, infection defense, and fish species growth, and it also identifies the routes via which it enters the aquatic environment. Dietary Se is important in fish feed due to its additive, antioxidant, and enzyme properties, which aid in various biological processes. However, excessive intake of it may harm aquatic ecosystems and potentially disrupt the food chain. This review explores the diverse natures of dietary Se, their impact on fish species, and the biological methods for eliminating excesses in aquatic environments. Soil has a potential role in the distribution of Se through erosion from agricultural, industrial, and mine sites. The research on dietary Se’s effects on fish immune system and growth can provide knowledge regarding fish health, fish farming strategies, and the health of aquatic ecosystems, promoting the feed industry and sustainable aquaculture. This review provides data and references from various research studies on managing Se levels in aquatic ecosystems, promoting fish conservation, and utilizing Se in farmed fish diets.

## 1. Introduction

Dietary Se is vital for the growth and immune system of fish. Dietary Se is a micronutrient that functions as a component of important selenoproteins for both humans and animals [1,2]. These selenoproteins are involved in various biological processes, including antioxidant defense, thyroid hormone metabolism, and immune function [3]. Se as a nutritional element is obtained in the aquatic environment from two main sources: feed and fertilizer. In areas where waste and other activities from agriculture, industries, mining, and other natural events are carried out, Se concentrations might seem higher because of closer point sources. However, human activities can transfer Se into aquatic food webs through pathway concentrations in seleniferous environments [4,5,6]. Increases in dietary Se can, however, upset the aquatic food chain, and overexposure may cause disruption in fish tissues, which is harmful to human consumption both directly and indirectly.

Aquatic animals may be subjected to biotic and abiotic stresses during the farming season [7]. The most important factor that helps to counteract these stressors and promote high productivity and well-being is a nutritionally balanced aquatic diet [8]. Another important measure to ensure the nutritional value of aquafeeds is the addition of microelements such as Se [9]. Se aids many biological functions in the bodies of animals, including those of an antioxidant, a metabolic stimulator, and an immunostimulant [8]. In fact, Se serves as a precursor for a number of metabolites involved in biological processes in the body [10]. The immune and antioxidant systems benefit from the role of Se in the synthesis of selenoproteins [11]. Moreover, it helps regulate liver and kidney functions as crucial organs for the release of body toxins and nitrogen residues [12]. According to Swain et al. [13], Se is crucial for the biological functions of the fish’s body. Nutrition is one of several elements that influence the growth and development of fish in aquatic environments. Selenium is a crucial dietary component that has a tremendous impact on fish growth. Understanding the effects of dietary selenium on fish growth is critical for sustainable aquaculture practices and maintaining ecological balance in aquatic ecosystems. Additionally, several fish species have been found to have better overall growth when their diets contain adequate levels of selenium. According to Wang et al. [14], adequate selenium supplementation in feed has been associated with improved feed efficiency, weight gain, and growth rates.

Additionally, dietary Se enhances immune responses and increases the production of immune-related molecules, such as interleukin-10 (IL-10) and immunoglobulin A (IgA), which are critical for immune regulation and defense against infection. Zhai et al. [15] found that Se supplementation has a positive effect on the immune system and intestinal barrier, reducing permeability and pathogen entry. Se also affects the function of various immune cells in fish, such as macrophages and lymphocytes, as it enhances the phagocytic activity of macrophages so that they can effectively engulf and eliminate pathogens [16]. It also modulates the production of pro-inflammatory cytokines, such as interleukin-1 beta (IL-1β) and tumor necrosis factor-alpha (TNF-α), which are involved in the immune response against pathogens [17]. In addition, Se enhances lymphocyte proliferation and antibody production, promoting a robust immune response in fish. Se has been found to regulate the expression of immune-related genes in fish. For example, studies have shown that dietary Se supplementation promotes the expression of genes involved in the immune response, such as Toll-like receptors (TLRs) and cytokines [14]. These immune genes are needed for both the identification of pathogens and the start of immune reactions. Rainbow trout fed abundant organic Se (Se-Plex^®^) strengthened their immunity with 4 mg/kg of dietary Se which triggered a crucial response in the head kidney (HK) of rainbow trout and upregulated genes [18]. Rathore et al. [19] conducted a study on nano-Se supplementation in the diet of Nile tilapia, focusing on Se assimilation and the expression of immunoregulated selenoproteins, which showed enhanced Se absorption and utilization, while altered expression levels of immunoregulated selenoproteins were observed, suggesting possible effects on the immune system. In another study by Fontagné-Dicharry et al. [20], dietary Se was found to significantly affect the antioxidant status of rainbow trout fry and parameters related to oxidative stress. Generally, fish feed meal is a vital source of dietary Se in the aquatic environment that helps strengthen stress, inflammation, and infection response [21].

Se in fish feed should be used in the correct dosage, which should not exceed the levels recommended by the European Union. According to previous studies, the recommended Se dose in fish feed falls between 0.1 and 2.00 mg/kg; anything above this can have negative consequences [22]. Also, the European Union has enacted legislation stating that Se supplementation with organic Se should not exceed 0.5 mg/kg in feed [23,24,25].

Furthermore, the insight gained from knowledge about Se contents in fish feeds may ultimately lead to maximizing profitability for the feed industry by increasing fish farmers’ confidence and the demand for fish feed meal [26]. Adequate Se-rich feed from the feed industry supports the fish immune system, prevents disease, promotes fast growth, and ensures high meat quality [27,28]. By providing Se-enriched feeds, the feed industry can help prevent disease outbreaks and promote fish welfare. The feed industry can contribute to Se biofortification, i.e., increasing the Se concentrations in fish products for human consumption. Se-enriched feeds can help address Se deficiencies in regions where soils are naturally low in Se [29]. This ensures that fish farmers have access to Se-rich fish feed, which benefits their health and well-being. According to Wang et al. [26], Se can guarantee better fish health and secure food production, even though several countries have implemented limitations on the Se concentration of fish feed. The feed industry must adhere to Se contents based on fish species acceptability and standards to improve reputation, trust, and profitability. Se is essential for fish nutrition, health, disease prevention, biofortification, and regulatory compliance, enhancing the industry’s overall performance.

This study aims to explore the biological effects of extreme dietary Se on fish species, its implications for various dosage levels, and preventative methods for excess Se. It helps fill the gaps in the existing literature and provides a comprehensive insight into the effects of Se on fish growth and immune systems.

## 2. Sources of Se and Their Effects on the Aquatic Environment

Se is a naturally occurring element that interacts with fish’s environment through anthropogenic and natural means (see Figure 1). Thus, Se concentration in soil and aquatic environments depends on anthropogenic activities (industrial, mining, agriculture inputs, sewage sludge, oils, power plants, and mines) and natural occurrences (weathering rocks, volcanic eruptions, and atmospheric processes). There are ways in which Se can directly and indirectly get into the soil and aquatic environment, including erosion, runoff, and leaching that release coal ash into water bodies, particularly in Se-rich regions [4,30]. However, it also gets into the aquatic environment directly from mine discharges, agricultural fields, waste water, sewage disposals, and industrial discharges [6]. Khoshnood [31] highlighted the most frequent contaminants, such as heavy metals and pesticides, causing harm to fish in aquatic environments. According to Tan et al. [32], 40% of Se in the atmosphere and aquatic environment comes from industrial and mining activities. This percentage of Se was primarily influenced by human activities like burning coal and crude oil, which serves as an influencing factor [32]. In addition, these activities by humans have introduced high Se levels as micronutrient elements to the environment, causing imbalances in the food chain. The level of Se concentration in the aquatic environment in a particular area depends on a number of factors, such as the increase in demand for agriculture, industrial processes, mining power plants, chemical wastes deposited in the environment, from which Se comes from different sources, and the release of pollutants into the aquatic system [33,34]. Fish in the aquatic environment perform all metabolism processes, including Se intake, which may be carried out orally or assimilated into muscle and liver tissues.

Fish species make excellent models for detecting genotoxic chemicals in aquatic ecosystems [36,37] due to their abundance and specific habitats [38]. Most toxicity levels are based on uptake in the aquatic environment, which is substantially lower in natural populations than uptake through the food chain. Se concentrations in soils vary depending on the location, activities, type, and nature of the soils. Tan et al. [39] reported that Se content in various soil types varies from 1.07 to 6.69%. Thus, natural processes (physical and mechanical) occurring in the soil could lead to an uneven distribution of Se in the environment. In addition, the Se content in the soils of many countries varies. Studies have found Se levels of 0.4 mg/kg in most parts of the world’s soil, including North America, Ireland, Australia, and Israel, as well as in some Chinese provinces (72% of the area of China), New Zealand, and a considerable part of Europe [40,41] (Table 1).

## 3. Forms of Dietary Se Intake and Their Effects on Fish Species

Fish species absorb dietary Se in various forms, such as organic and inorganic [45,46,47]. These forms of dietary Se impact fish’s biological functions. The forms of dietary Se aid fish species in several ways depending on their health, size, species, feed composition, and experiment conditions [48]. The biological functions of fish in their aquatic environment may be supported by dietary Se use. However, the majority of dietary Se absorption by fish species is efficient due to their bulkiness or minute size [49].

As for Se in the diet of fish, there are a few different forms commonly used as inorganic, such as Na_2_SeO_3_ or sodium selenate (Na_2_SeO_4_) (Table 2). They differ in toxicity, with Na_2_SeO_3_ being slightly less toxic than Na_2_SeO_4_ [50]. Na_2_SeO_3_ is commonly used in fish feed and is also found in plant foods, meats, seafood, and dietary supplements that provide numerous health benefits, while Na_2_SeO_4_ is an insecticide used in agriculture to control insects and can be used as a fungicide [51]. Both forms are readily absorbed by fish and can meet their Se requirements. Na_2_SeO_3_ plays an important role in fish nutrition and tissues, as a study by Rathore et al. [19] showed that supplementing fish feed with inorganic Se positively affects Se content in Nile tilapia tissues.

Organic forms of dietary Se also include selenomethionine (SeMet) and selenocysteine (SeCyst). The two forms often come from natural sources such as yeast or plants. According to a study by Penglase et al. [52], fish feed enriched with Se provides fish with a higher proportion of essential nutrients and thus promotes their accumulation in their tissues. Additionally, dietary organic Se in fish feed has resulted in strengthened retention and increased Se levels in the fish. Since fish tissues accumulate both essential and non-essential trace elements [53], it is important to have an understanding of the various Se forms and their effects on fish.

SeMet is one of the essential amino acids. Fish can obtain SeMet from their diet, and it is known to play a crucial role in their overall health and biological functions such as growth and the immune system. A study by Liu et al. [54] demonstrated that SeMet supplementation in grass carp diets contributed to fostering growth performance, immune response, and antioxidant capacity.

SeCyst is an amino acid that incorporates Se into selenoproteins, thereby affecting reproduction, egg quality, and spawning performance in fish [26]. It is important for protecting fish from oxidative stress and maintaining cellular health, as shown by Kumar et al. [55]. In addition, SeCyst is an important component of antioxidant enzymes such as GPxs.

**Table 2 animals-13-02978-t002:** Forms of dietary Se and their effects on some fish species.

Forms of Dietary Se Intake	Effect	Refs.
**Na_2_SeO_3_**	Its supplementation in Mahseer fish shows better growth and biochemical health.	[56]
	It enhances African catfish growth.	[57,58]
	Na_2_SeO_3_, a Se dietary supplement, improves fish growth, antioxidant capacity, and immune response in fish, including Nile tilapia.	[59]
	It supports the growth and survival rates of rainbow trout.	[60]
**SeMet**	It supports better absorption and more bioavailability in species such as rainbow trout, fathead minnows, channel catfish, and Atlantic salmon.	[61]
	It positively enhances innate immunity by improving serum lysozyme activity and GPxs activities in the liver and muscle tissues of Mahseer fish.	
	It causes better protective influences during stressful conditions with Medaka.	[56]
	Se content elevates the healthy growth of common carp.	[26]
	At 0.6 mg/kg, it supports the growth and immunity of goldfish.	[57]
	It stimulates the larval growth of rainbow trout fish.	[62]
	It promotes the growth potential of African catfish.	[58]
**Se Nanoparticles (nSe)**	It enhances growth and improves antioxidant enzyme activities in juvenile black sea bream.	[14]
	It increases the growth and immunity of goldfish at 0.6 mg/kg.	[57]
	It can alleviate hepatopancreas injury in high-fat diets and enhance the survival of grass carp.	[63]
**Na_2_SeO_4_**	It enhances the growth performance of African catfish.	[58]

## 4. Se Characteristics in the Soil and Their Influences on the Aquatic Environment

Se is a chemical element that can naturally occur in soil. Soil serves as the primary source of Se, influencing its content in food, vegetables, and drinking water [39]. Hence, the amount of Se in the soil directly influences the amount of Se in fish. Soil parent material, physical and chemical properties, and other factors like soil type and land use could influence Se distribution in soil [64]. Se distribution in soil varies geographically and depends on factors such as parent material, climate, and land use practices. It can be found in varying concentrations worldwide (see Table 3), with higher levels typically found in regions with seleniferous deposits [64]. The bioavailability of Se in soil depends on its chemical form, as it can exist in different oxidation states (−2 to +6), with selenate (SeO_4_^−^) and selenite (SeO_3_^−^) being the most common forms. Na_2_SeO_3_ is more mobile and bioavailable, while SeO_4_^−^ is prone to adsorption and immobilization in soil [65]. Soil contains various sources and forms of Se, including Se sulfides and Se ions. It is acidic, oily, and rich in organic matter. Se exists in various organic and inorganic forms, with selenopyrite or selenopyrite fixed to organic matter. According to Saha et al. [44], dietary Se exists in selenide (Se^2−^), SeO_3_^2−^, and SeO_4_^2−^ forms, found in acidic, low redox soils, predominant in acidic, medium redox soils (0 to 200 mV), and dominant in alkaline, high redox soils (500 mV). Moreover, soil pH, organic matter content, and redox conditions significantly influence Se concentrations in soil. Acidic soils with low organic matter content can increase Se availability, while alkaline or high organic soils may reduce its mobility [66]. The interaction of Se in soil can have impacts on the aquatic environment, including fish populations. When Se-rich soils are eroded or irrigation water containing Se is used, Se can enter nearby water bodies through runoff or leaching. This can result in elevated Se concentrations in aquatic environments [67].

In the soil environment, the ideal Se levels, which support a variety of living organisms, including some fish species, particularly the African catfish in terms of extreme conditions, some crustaceans like crab, lobster, and prawns, as well as mollusks like snails and slugs, should exist. The soil environment serves in the capacity of producing food, transferring energy within food webs, releasing water, and regulating carbon [26,68]. Living organisms, including fish in the soil and aquatic environment, require adequate Se based on their acceptability for their survival and welfare. However, high Se concentration depends on soil types and characteristics such as sedimentary and organic matter, whereas magmatic rocks are poor in Se concentration and have a low Se content [69]. In areas where aquaculture structures are unmodernized, they allow external sources of Se to enter aquatic environments through runoff.

Moreover, authors have reported the global average Se concentration in soils at 0.1 and 2 mg/kg [70], 0.01–0.4 mg/kg [71], and 0.03–2.0 mg kg^−1^, with an average of 0.40 mg/kg [71,72]. The soil has multiple Se levels, and its characteristics are high, toxic, deficient, marginal, and moderate in levels of 0.40–1.00 mg/kg, 1.00–3.00 mg/kg, >3.00 mg/kg, <0.125 mg/kg, and 0.125–0.175 mg/kg [73], respectively, depending on type and characteristics. However, the average Se concentration level in soil globally falls within the range of 0.01–0.4 mg/kg [74,75]. The levels of Se in the soil vary based on the region or countries, as indicated in Table 3.

**Table 3 animals-13-02978-t003:** Se concentration in soil environment and its characteristics.

Countries	Se Concentrations in Soils (mg/kg)	Refs.
Belgium	It has low Se content in agricultural soils, with an average of 0.30.	[72,76]
China	Its average Se range falls within 0.01–2-Se and most top soils contain 0.06–9.13	[65,77,78]
Denmark	A deficient soil Se level was found to be between 0.1 and 0.6.	[67]
England	<0.01–16	[67]
Global Se levels	The average Se range in the world falls between 0.01 and 2.00.	[4,67]
Iran	0.04–0.45	[79]
Japan	It has an average Se range in the soil between 0.05 and 2.80.	[71,72]
Netherlands	Aqua regia has average Se contents of 0.12–1.9 for grassland soils and 0.20–1.20 for arable land soils.	[72]
Qatar	0.12–0.77	[80]
Scotland	It has a general average Se level in most soils between 0.12 and 0.88, and sandy soil contains 0.43.	[67,81]
Sweden	It has a low average Se content in agricultural soils of 0.30, the same as Belgium.	[72,82]

## 5. Se Influences the Biological Functions of Various Diets of Fish Species

Many fish species’ biological processes are positively impacted by dietary Se, including thyroid regulation, growth promotion, improvement of fertility, immune system enhancement, and delay of aging [83,84,85], as well as antioxidant activity [86,87]. In contrast, Mechlaoui et al. [88] discovered that gilthead sea bream fed SeMet had better growth performance and higher antioxidant content than those fed sodium selenite (Na_2_SeO_3_). Ma et al. [89] demonstrated that dietary Se supplementation resulted in improved growth of grass carp. Abdel-Tawwab et al. [58] studied African catfish fed diets containing 0.2, 0.4, or 0.6 mg/kg Se and were also exposed to copper toxicity. They found that organic Se improved growth and reduced the negative effects of copper toxicity on physiology. Elia et al. [90] confirmed that fish fed Se-supplemented diets had higher growth rates and Se accumulation than the control group. Zhu et al. [91] investigated the effect of dietary Se on growth, body composition, and hepatic GPxs activity of largemouth bass; the researchers discovered that dietary Se supplementation improved growth and body composition and increased hepatic GPxs activity in largemouth bass, which may have a positive effect on their health. Han et al. [92] found that higher dietary Se content positively affected the overall growth performance of gibel carp compared to diets with lower Se content. Saffari et al. [93] confirmed that dietary Se is important to improve the growth and productivity of common carp and Ashouri et al. [94] reported that dietary Se concentrations of 0.1, 0.2, and 0.4 mg/kg improved the growth of common carp, muscle composition, and antioxidant status. Ibrahim et al. [8] confirmed that Nile tilapia fed nanoselenium (nanoSe) supplements had better growth rates and feed efficiency than those fed Se bulk supplements. Se is also important for biological functions like promoting fish growth, immune function, and health. It also boosts fish immunity, antioxidation, the stabilization of cell membranes [95,96], and the maintenance of normal metabolism in the body [30]. Fish feeding on regulated Se may support fish species performance by contributing to weight gain and specific growth rates in aquatic animals [64,93,94]. Studies on Wuchang bream have shown that Se increases growth hormone (GH) and insulin-like growth factor (IGF)-I levels in serum and messenger ribonucleic acid (mRNA) transcription levels of growth hormone receptor 2 (ghr2) and insulin-like growth factor 1 (igf1) in the liver [97]. It effectively promotes growth in various fish species, including grass carp, gibel carp, and rainbow trout [92,98].

## 6. Excessive Se Exposure to Fish and the Types of Deformities or Abnormalities Experienced

Malformations or abnormalities are extreme conditions that no fish farmer likes to experience. Most problems that occur in fish farming are due to poor management and feed is a major factor. Deformities or abnormalities in fish can be caused by several factors. Dietary Se is required by organisms in sufficient amounts for proper biological and physiological functioning. However, increased dietary intake of Se may lead to adverse effects including malformations in fish. A study conducted by Lemly and Smith [99] examined the effects of dietary Se on fish deformities. They found that high dietary Se significantly increased the prevalence of dorsal deformities in fish, such as lateral curvature of the spine. The deformities have been observed in both juvenile and adult fish, demonstrating the importance of monitoring Se levels in aquatic ecosystems. In addition, a research article by Lemly [35] examined the relationship between Se concentration and deformities in fish populations (see Table 4 and Figure 2). The study found that elevated Se concentrations in fish tissues were associated with an increased incidence of skeletal deformities, including gill malformations and spinal anomalies. It should be noted that Se can enter aquatic ecosystems through natural sources, such as rocks and soils, as well as anthropogenic activities, such as mining and industrial effluents. Therefore, it is critical to carefully control Se levels in aquatic environments to avoid adverse effects on fish populations. Excessive dietary Se intake can interfere with the normal development of fish embryos and larvae. In a study by Johnson et al. [100], elevated Se exposure early in life was shown to increase the incidence of cranial deformities in fish. High dietary Se concentrations may also affect fish reproduction. Research by Lemly [35] has shown that Se accumulation in fish tissues can lead to reduced egg viability and offspring survival. In addition, Se-induced reproductive impairments can result in reduced population growth and overall reproductive success in fish. However, higher Se intake can disrupt immune functions, making fish more susceptible to infection and disease. Studies by Lin et al. [101] and He et al. [102] found that elevated Se concentrations in fish tissue resulted in impaired immune responses and increased mortality due to infectious diseases. Dietary Se may also affect fish behavior. Dhara et al. [103] showed that Se exposure alters the swimming behavior and activity patterns of fish, potentially affecting their ability to forage for food, evade predators, and interact socially within their ecosystem.

Dietary Se as a nutritional element and its toxic concentration levels have drawn increasing attention to Se toxicity [104]. Moreover, Se is considered the second most harmful trace element in fish after mercury [45,105]. For these reasons, Se should be present in fish feed in appropriate amounts. As can be seen in Table 4, abnormalities in early-life stage fish [46] exposed to Se have been associated with negative effects on fish populations. Although there is evidence of Se toxicity in fish [106,107], it is not yet known what level is required to protect fish populations [106].

**Table 4 animals-13-02978-t004:** Deformities or abnormalities associated with Se toxicity in fish.

Abnormalities of Higher Se	Effects	Refs.
Renal calcinosis	A high Se content of 10 mg/kg causes renal calcinosis disorder in rainbow trout.	[108]
Metabolism disorder	A diet containing Se at 13–15 mg/kg was observed to be highly toxic for rainbow trout and channel catfish.	[108]
Oxidative stress and altered lipid metabolism	A high content of dietary Se in the Atlantic salmon diet at 15 mg/kg causes oxidative stress and altered lipid metabolism.	[109]
Loss of appetite and changes organs malfunctions	High dietary Se of 20.9 mg/kg in juvenile yellowtail kingfish causes reduced feed intake and liver and spleen changes.	[110]
Vertebral and muscle deformities	High Se concentrations cause spinal column abnormalities in fish, such as curvature or fusion of vertebrae, and hinder muscle growth.	[110,111,112,113,114]
Survival abnormalities	Razorback sucker larvae’s survival is reduced by Se concentrations above 4.6 mg/kg.	[2]
Fin deformities	Se toxicity affects fish fin development (short, frayed, or missing fins), causing deformities.	[51]
Eye abnormalities	High Se levels have been linked with fish eye deformities (reduced eye size or even complete eye loss) and cataracts.	[51]
Gill deformities	Se can affect the delicate structures of fish gills, potentially leading to abnormalities like fused or malformed gill arches, reduced gill surface area, or impaired respiration.	[115]
Reproductive abnormalities	Se can impact fish reproductive systems, causing deformities in gonads or reduced fertility.	[52]
Abnormalities of liver, gill cells, and nerve conduction	Extreme Se intake causes abnormalities in the liver and gill cells, nerve conduction disorders, and defects in the energy metabolism of zebrafish.	[111]
Craniofacial deformities	Se toxicity can affect the development of the skull and facial structures in fish, resulting in malformations such as asymmetrical jaws or abnormal eye placement.	[109]
Skeletal deformities	Extreme Se exposure may disrupt bone growth and mineralization in fish, leading to skeletal deformities such as bent or twisted fins, abnormal body shapes, or curved spines.	[111]
Larval deformities	Se accumulation can result in northern pike larvae deformities, including spinal curvature, craniofacial malformation, and impaired swimming ability.	[113]

**Figure 2 animals-13-02978-f002:**
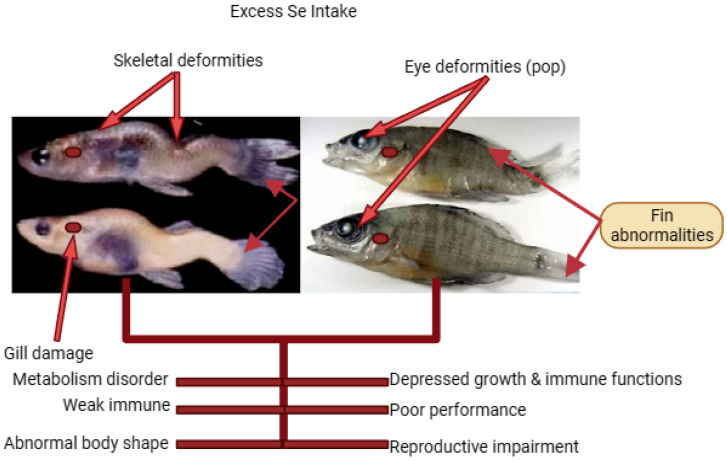
Increased Se impact on fish. The excess Se impact on fish was retrieved from Lemly [35], Wang et al. [26] Khan et al. [61].

## 7. Dietary Se Deficiency in Fish Feed and Its Biological Effects

Dietary Se is a crucial component for the healthy growth and immune function of fish. It serves as a cofactor for crucial enzymes involved in immune system control, thyroid hormone metabolism, and antioxidant defense [89]. Dietary Se deficiency and the ensuing health issues may develop if fish do not consume sufficient Se in their diets. However, fish farmers may start experiencing impaired growth as one of the main effects of dietary Se deficiency in fish. According to studies [113], it has also been linked to reduced feed conversion, decreased protein synthesis, and poor tissue development. Aquaculture businesses may ultimately suffer from stunted growth and subpar performance as a result of this deficiency. Additionally, fish have reproductive capabilities, including gonad growth, maturation, and effective spawning [113,114,115].

Consequently, these processes can be disturbed by insufficient dietary Se in fish feed (see Table 5), which can also result in decreased fecundity and poor egg quality. Fish with low Se levels also have weakened immune systems and are more susceptible to illness and infection. By promoting the development of immune cells and controlling inflammatory processes, Se is known to enhance immunological responses [116]. These defenses are weakened by insufficient Se consumption, leaving fish more susceptible to infections and jeopardizing their general health. In addition to dietary Se deficiency in fish feed, the aquaculture feed industry should employ different approaches to address this problem. These approaches include adding Se-enriched feed ingredients like plants, animals, and yeast as organic dietary components [26]. These approaches may therefore be employed to correct dietary Se deficiency in order to enhance fish health, growth, and performance.

## 8. Biological Effects of Dietary Se on Fish Species

The presence of Se in the fish diet increases respiratory activity, lysozyme activity, acetylcholinesterase, and myeloperoxidase activities. In a recent report by Liu et al. [64], they discovered that Se increased the genes for fatty acid oxidation and triglyceride hydrolysis in the liver. The biological processes of aquatic animals are more influenced than negatively affected by the presence of Se in their diet. Se performs a variety of functions in the internal processes of aquatic animals, including immune system development, nutrition and nutritional conditions, enhancing reproduction, and increasing growth and performance in the aquatic environment.

The growth and performance of fish in aquatic environments depend on many factors, such as water quality, feed, and other genetically determined factors. Feed is one of the factors that should be balanced, with Se being an important nutrient for the biological activities of fish. Notwithstanding the effectiveness of Se in fish feed, serving as a vital nutrient requirement for fish growth, it has also been associated with improved feed digestibility, health, and growth in finfish. It has equally been associated with the stimulation of growth hormone production, leading to high growth [19], binding to the enzyme deiodinase necessary for thyroid hormone regulation [11], and an increase in growth hormone [10].

As reported by authors, Se is a vital requirement for certain fish species in improving their growth, feed utilization, and immune resistance to diseases of Mahseer [50,120], Asian seabass [120], Red seabream [121], Nile tilapia [122], Goldfish [57], Gilthead sea bream [120], Mahseer fish [61], European seabass [123], Rohu [13], and Caspian roach [124], as seen in Table 6.

## 9. Effects of Se Toxicity Intake on Immune Genes in Fish

Toxicity levels in fish feed vary based on species. Some fish species grow and survive above the European Commission’s recommended levels of Se and the optimum Se content range for aquatic animals of 0.2–1.8 mg/kg diet [129]. Every fish species has its own tolerance for dietary Se intake. However, one species’ acceptable level could be harmful to the other species’ biological effects, including growth performance and immune functions in the aquatic environment. For example, white sturgeon are less sensitive to Se toxicity, with a dietary Se toxicity level of 10 mg/kg [130], while other fish species are highly sensitive to it at these levels. Also, despite the beneficial effects of dietary Se on fish, it has been reported that excessive intake of Se can reduce antioxidant capacity and growth performance [6,131,132]. The severe Se intake in fish feed may be hazardous to certain fish species, thus causing biological abnormalities like reproductive impairments, egg and larval transfer, yolk sac absorption, reduced hatching, teratogenicity (deformities), and oedema in the early-life stages of fish and dying larvae [45,51]. Se is an essential element for upregulating and downregulating genes, but its deficiency can lead to the downregulation of antioxidant defenses and thyroid hormone metabolism, negatively affecting fish health [45,51]. Also, its excess may cause the upregulation of antioxidant system genes, which benefits fish by combating oxidative stress and maintaining cellular health [133].

However, dietary Se is essential for optimal growth, metabolic processes, performance, immunity, disease resistance, and reproduction in fish, but its deficiency poses a threat to their survival. Authors have reported that extreme Se levels have adverse effects on various fish species, with highly toxic effects at 13–15 mg/kg in rainbow trout [108,134], adverse health and survival effects at 6.6 mg/kg in juvenile split tail [135], reduced growth (20–30 mg/kg) in fathead minnow [59], and toxic levels at 10 mg/kg in white sturgeon [130]. Extreme Se levels in fish feed have been reported to cause locomotive neuron disorders in zebrafish. According to Zhao et al. [112], excess Se in zebrafish embryos can cause motor neuron dysfunction by inhibiting sl1 and insml1a expression, and also disrupting gamma aminobutyric acid (GABAergic) neuron function, excitability, and differentiation of dopaminergic neurons by downregulating glutamate decarboxylases (gad1b and gad2) expression, causing reduced spontaneous motor activity and affecting dopaminergic neuron differentiation [135,136]. Watanabe et al. [137] and Lemly [35] reported that higher Se can cause teratogenic abnormalities in many organs of fish, including the spine, head, mouth, and fins, as well as reduced feed efficiency, tissue breakdown, and reproductive failure. Also, Cotter et al. [138] reported that Se deficiency in fish diets leads to growth retardation and immunological instability in fish. Se acts as an antioxidant, protecting cells from oxidative damage and supporting immune cell function.

## 10. Biological Methods of Se in Controlling Aquatic Environments

Fish need Se to maintain their good health and biological processes; hence, it is necessary for Se to be present in aquatic environments. However, contaminants in the aquatic environment have been a concern, as man’s activities in securing a better life have also been detrimental to the environment. Many bacteria and plants have been used to treat or remove Se concentrations in the aquatic environment, and some of these methods and techniques are eco-friendly. However, researchers have found that microbial species can convert soluble Se (Na_2_SeO_3_ and Na_2_SeO_4_) to Se^0^ [139,140].

In aquaculture, several preventative or treatment techniques have the benefit of being toxic-residue-free pollution treatments that can be recycled and reused in the aquatic environment [141,142]. Trace metals are high-level pollutants in the aquatic environment, especially Se that can be prevented or treated using different techniques like bioremediation, adsorption, phytoremediation, constructed wetlands, nutrient management, etc. Toxic pollutants can also be removed using various biological techniques, or bioremediation, which uses plants and microorganisms. With regard to biological processes, including gene expression for the immune system and the growth performance of aquatic animals that are extremely sensitive to Se, this can be summarized in Table 7.

Regardless of the methods used to control excess Se concentrations in soils and waters, several factors must be considered, such as the specific environment, the severity of the contamination, and the desired outcome. However, one method that is generally considered effective is phytoremediation. It is considered efficient in removing Se from soil and water [143]. In this method, plants are used to remove or absorb Se. For example, hyperaccumulator plant species such as garlic and cabbage plants, such as cabbage, mustard, and broccoli, are able to accumulate high Se concentrations in their tissues, effectively lowering contamination levels [144]. It is also considered cost-effective compared to other methods such as constructed wetlands, phytovolatilization, bioremediation, etc. Once established, plants can take up and accumulate Se naturally, requiring minimal maintenance and avoiding the need for expensive infrastructure. It is also considered sustainable and environmentally friendly compared to other Se control methods [145]. It relies on natural processes and interactions between plants and microbes to remove Se and minimizes the use of chemicals or disruptive techniques [146]. In addition, once Se accumulates in plant tissues, it can be harvested and properly disposed of, preventing the re-release of the contaminant into the environment. This ensures long-term stability and reduces the risk of recontamination. Phytoremediation is the preferred method, but suitability depends on site-specific factors.

**Table 7 animals-13-02978-t007:** Biological methods of Se prevention and its effects.

Biological Control Methods	Approaches	Refs.
Phytoremediation	Plants can accumulate Se in aquatic environments, potentially aiding in reducing Se in contaminated ecosystems.	[147,148]
	It has the ability to absorb contaminants through plants’ roots and breakdown pollutants, especially Se, through bioaccumulation.	[149]
Bioremediation	This technique uses microorganisms to break down and transform Se compounds, enabling bioremediation of Se-contaminated water through the use of bacteria, fungi, and algae.	[144,149]
Adsorption	It involves the applicable removal of Se in the aquatic environment as it has a low operating cost, high Se tolerance, and is effective in aquatic environments.	[150,151]
Constructed wetlands	Engineered wetlands mimic natural wetlands by using plant species and microorganisms to remove pollutants, including Se, from contaminated water sources.	[143]
Bioaugmentation	This is the process of introducing Se-reducing microorganisms to aquatic environments for natural remediation, transforming toxic Se into less harmful forms.	[149]
Nutrient management	Effective nutrient management in aquatic ecosystems can indirectly reduce Se toxicity by reducing agricultural runoff and wastewater discharge.	[149]
Phytovolatilization	This is an indirect method of removing contaminants or pollutants through absorption by plants’ roots in the soil and releasing them into volatile contaminant flux.	[151]

## 11. The Relevance of the Feed Industry in Monitoring Dietary Se in Farmed Fish

The feed industry takes seriously its responsibility to monitor Se levels in aquaculture and aquatic habitats. It uses a variety of approaches and procedures to ensure the safety and health of fish and the environment. The industry regularly samples and examines fish tissue to monitor Se levels in farmed fish. For this purpose, Se content is usually measured in the liver or muscle tissue of the fish. In this way, the industry can ensure that the Se content of the fish remains within acceptable limits [43,44,45]. The feed industry monitors not only the Se content in the fish but also the water conditions in which the fish are raised. They conduct thorough water quality tests, which include the Se content of the water. This helps them assess the overall health of the environment and identify potential threats to the fish [35]. The feed industry often collaborates with regulatory agencies and academic institutions to ensure proper monitoring. This collaboration enables it to keep up with the most recent scientific advancements, recommendations, and legislation related to Se monitoring. In addition, the feed industry may take the necessary steps to maintain a safe and healthy fish habitat and limit any potential impact on the ecosystem by actively monitoring Se levels in both farmed fish and aquatic ecosystems.

## 12. Conclusions

Se is essential for numerous biological processes and at an acceptable dose, for the capacity of fish to survive, despite its harmful effects at high concentrations. Se is ingested by fish species through a variety of sources and processes. The dietary Se found in the aquatic environment is derived from runoff and leaching; it can also exist in a variety of forms, however, it is more vital for fish existence, growth, and gene and gene-related functions. Its toxicity depends on the dosage ingested and fish can absorb it through a variety of organs through the bioaccumulation process. Also, Se is necessary for the development, defense, procreation, and enzymatic functions of fish. Hence, excess intake can lead to deformities and other abnormalities, but it is important for their daily biological functions. Furthermore, Se intake by fish in aquaculture has an acceptable optimal range in which extremes are considered high or low concentrations. The environment plays an important role in the availability of Se due to natural and anthropogenic activities and several influencing factors. In the world, every country has its own Se range in the soil and this has a direct or indirect relationship with the aquatic environment. Dietary Se is vital for the growth and immunity of certain fish, with different dietary acceptability. It also plays an important role in the biological functions of fish, including increasing respiratory burst, lysozyme, acetylcholine esterase activity, and myeloperoxidase activities. However, excess Se in aquatic systems can be prevented by various biological methods or techniques.

## Figures and Tables

**Figure 1 animals-13-02978-f001:**
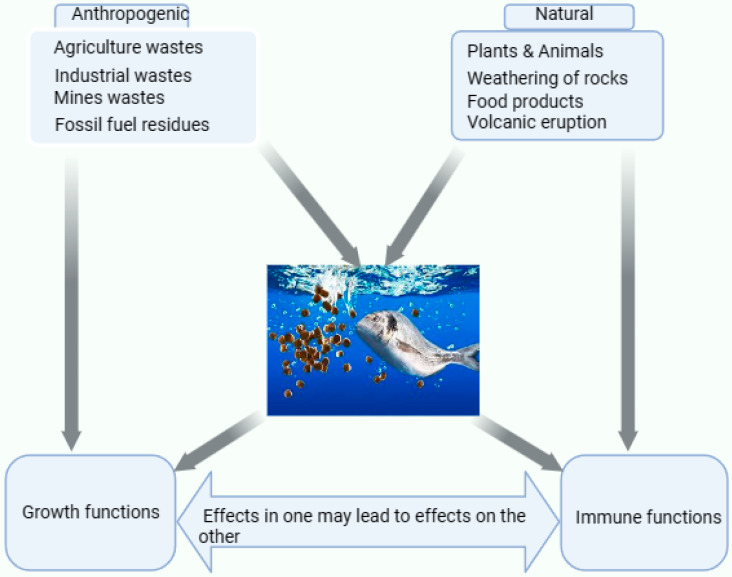
The sources of Se and their influences in the aquatic environment with the fish. Se in the aquatic environment may be derived from artificial (industrial wastes, fossil fuel residues, and mines) and natural sources (plant and animal products, weathering of rocks, volcanic eruptions, and food products). The tissues responsible for Se intake include the gills, muscles, liver, and kidney [35]. The Se obtained in the aquatic environment could either influence their biological functions, such as growth and immunity, which have an inverse relationship wherein the impact on growth will influence immunity, or vice versa.

**Table 1 animals-13-02978-t001:** Summarizing the sources and effects of the dispensation of Se in the aquatic environment.

Sources of Se	Dispensation of Se in the Aquatic Environment	Effects of Action	Refs.
Agriculture	Derived from mining sites, fertilizer application, and sewage sludge (municipal).	It enhances Se in the aquatic environment through surface runoff, infiltration, leaching, and percolation.	[42,43]
Natural activities of parent rocks	Parents’ rocks weathering contributes to Se presence in environment.	Rocks contain various metals, and Se is obtained from their weathering and gets into the surface soils and groundwater of the aquatic environment	[42,43]
Industrial activities	Crude oil, coal, fossil fuels, nuclear fuel, mining, and smelting processes elevate Se levels in the soil and aquatic environment.	Several industries are located along aquatic environments for water and in most cases, dumping can easily occur.	[41,44]

**Table 5 animals-13-02978-t005:** Dietary Se deficiency and its body system effects on fish.

Se Deficiency in Feed	Effects	Refs.
Growth and feed utilization	Biological functions, as well as the regulation of selenoproteins.	[117]
	Regular release of thyroid hormones, which stimulate growth hormone secretion from the pituitary gland of fish.	[61]
	Enhances antioxidant enzyme activities (e.g., catalase (CAT), Glutathione reductase, and GPx), reduces malondialdehyde (MDA) content, protects the liver from reactive oxygen species (ROS), maintains lipid metabolism, and protects the liver from ROS threats from nitrite exposure.	[91,118]
	Eliminates hydrogen peroxides and scavenges lipid peroxidation, especially as a selenoprotein GPxs active center.	[61]
	Enhances antioxidant enzyme activities (e.g., catalase (CAT), Glutathione reductase, and GPx), reduces malondialdehyde (MDA) content, protects the liver from reactive oxygen species (ROS), maintains lipid metabolism, and protects the liver from ROS threats from nitrite exposure.	[91,118]
	Innate immune response through selenoproteins under stress conditions.	[118]
	Help animals counteract oxidative stress to protect cellular membranes.	[118]
Growth and immune	Poor growth performance, lipid peroxidation, and decreased GPx enzymatic activity are observed in most fish due to Se deficiency in African catfish, Prussian carp and Japanese abalone.	[61,118]
	Inadequate supply of Se causes lower growth and a decreased GPxs activity in channel catfish.	[119]
Disease invasion	Salmon showed susceptibility to the Vibrio pathogen (Hitra disease).	[22]
Nutritional imbalance of Se	A decrease in plasma GPxs activity in rainbow trout occurs when fed a Se-deficient diet.	[119]
	Reduced GPxs activity in the rainbow trout liver.	[22]
	A decrease in GPxs activity in the liver and muscle tissues.	[57]

**Table 6 animals-13-02978-t006:** Dietary Se effect on growth, immune responses to diseases, and feed utilization of fish species.

Species	Effects	Refs.
Gilthead seabream	It helps in improving the growth of the species larvae and preventing skeleton anomalies.	[125]
Asian seabass	It aids growth performance and immune response, and decreases alanine aminotransferase (ALT) and aspartate transaminase (AST) levels with no significant change in liver superoxide dismutase (SOD), GPx, or CAT.	[119]
Red seabream	It supports growth, feed efficiency, protease activity, hematocrit, antioxidant potential, and significantly decreases reactive oxygen metabolites, cholesterol, and triglycerides.	[121]
	It boosts the tolerance due to the stress caused by low salinity with a diet containing 2 mg/kg of Se.	[121]
	It increases growth, feed utilization, GPx, SOD, CAT, nitrotetrazolium blue (NTB).	[121]
Nile tilapia	It increases growth, feed utilization, GPx, SOD, CAT, NTB, lysozyme, liver, TNF-α, IL-1β, MDA, HSP70, lysozyme, respiratory burst activities, and antioxidant enzymes and also serves as resistance against *Aeromonas sobria*.	[10,122]
	When given dietary Se at a dose of 1–2 mg/kg, the species’ resistance to *A. hydrophila* and *Aeromonas sobria* infection is increased.	[10,122]
European seabass	It encourages growth, feed efficiency, SOD, CAT, GH, IGF-1, IL-8, IL-1β, and lysozyme activities, as well as significantly reducing MDA and HSP70.	[122]
Mahseer fish	It raises serum growth hormone levels, lysozyme activity, and hematocrit, and increases growth of the species.	[61,126]
Sutchi catfish	It fosters growth, immunity, and resistance to *Aeromonas veronii biovar sobriam*.	[73]
	A 0.3 mg/kg dietary Se boosts resistance to an *A. hydrophila* infection.	[55]
Rainbow trout	It raises GPx without having a major impact on body composition, growth, survival, or the use of feed.	[124]
	When dietary Se is given at a rate of 2 mg/kg, it also strengthens their resistance to sublethal ammonia stress.	[124]
Caspian roach	It considerably promotes the ability to withstand malathion stress.	[57,123]
	It helps in relieving the impacts of malathion-induced stress with dietary Se of 1 mg/kg.	[57,123]
Gibel carp	It also fosters weight gain, specific growth rates (SGR), immunity, mucosal immunity, ghrelin, and IGF-1 expression while decreasing the feed convection ratio (FCR).	[127]
	It improves the GPx and quality of male sperm.	[127]
Common carp	It strengthens growth, liver GPx, SOD, and CAT activities, and also decreases MDA, AST, alanine transaminase, and lactate dehydrogenase.	[94]
Rohu	It lowers lactate dehydrogenase and alkaline phosphatase activities in *Aeromonas hydrophila* while enhancing growth and resistance.	[13]
Whiteleg shrimp	It boosts growth and decreases FCR significantly.	[128]

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
