# Peer review of "Effect of Dietary Selenium on the Growth and Immune Systems of Fish"

_animals, 2023, doi:10.3390/ani13182978_

Round 1

Reviewer 1 Report (Previous Reviewer 1)

The manuscript is a review on the effect of selenium on the growth and immune system of fishes. I find it useful for physiologists, zoologists, ecologists and geochemists.

The authors have significantly revised and improved their manuscript.

I have the following comments:

1) Figure 1. Looks bad for the reader. First, it is better to change the dark background to a light one. Secondly, small pictures inside the Figure are hard to see, despite the fact that they do not carry any information. It is better to remove these icons. It is better to present the scheme with blocks on a light background (white) in a large font and with straight arrows.

2) Table 1. I don't see references to sources in the second line "Natural activities of parent rocks".

3) Lines 165-167: why is the text in bold?

4) Table 2. Na2SeO3 is repeated in the table in two different rows, this does not make sense. In the table, it is better to describe the effect in the form “does something in some species of fish in some dosage”. Dosage indication is optional.

5) Line 223: remove the comma at the end of the sentence.

6) Line 231: which authors?

7) Table 3: what does «England/wales» mean? The United Kingdom is also marked at the end of the table. Scotland is also marked separately. I recommend compiling this, because It's about one island. In the same table, two ranges of values are indicated for Japan, what does this mean? The same goes for China. In general, I recommend that authors structure the table by continent rather than alphabetically. And specify the generalized global value either in the first or in the last line.

8) Figure 2. What does A and B mean? I didn't see the link to the picture in the text.

9) Table 4. There is no reference to the source in the line "Eye abnormalities".

10) Table 5. The column "Se deficiency in feed" contains either physiological processes or body systems. You need to rename this column. Or do the authors mean that the reduction in selenium consumption is associated with these processes? In any case, it will not be clear to the reader.

Some of the effects in the table are described in a very general way. What does “Maintain physiological health and growth”, “Biological functions” mean - these phrases do not carry any information. Two lines are missing a link to the source: "Eliminate hydrogen peroxides and scavenge lipid peroxidation, especially as a selenoprotein GPxs active center", "Poor growth performance, lipid peroxidation and decreased GPx enzymatic activity are...". For some reason, the last two lines are in bold and do not have a designation in the first column.

11) Table 6. As in other tables, not all rows (effects) have links to sources. In the table, it is better to describe the effect in the form “does something in some species of fish in some dosage”. Dosage indication is optional.

12) In the chapter Effects of Se intake on immune genes in fish we are talking about selenium toxicity, the chapter should be renamed.

13) Phrases like “Se presence in the aquatic environment is important for fish, which are significant for the health and biological functions of fish” are often repeated in the text, in conclusions and conclusions, about the same. I would reduce the number of such phrases in the text, it makes no sense to constantly repeat them.

14) There are phrases in bold in the text - what do they mean? It is also not clear what some terms mean, which the authors capitalize in the middle of the text - this has some special meaning (for example, line 421)

I believe that the article can be published after being improved and corrected.

Author Response

Please find attached document

Reviewer 2 Report (Previous Reviewer 4)

I congratulate the authors on the improvements to their work. In my opinion it can be accepted in its present form.

Author Response

Dear Editor,

Thank you for giving us the opportunity to submit a revised draft of our manuscript titled Effect of Dietary Selenium on the Growth and Immune Systems of Fish. The manuscript provides insight into the effects of dietary selenium on the growth and immune systems of fish (animals-2605274). We appreciate the time and effort that you and the reviewers have dedicated to providing valuable feedback on our manuscript. 

Regards,

Sahr Lamin Sumana.

Reviewer 3 Report (Previous Reviewer 5)

ABSTRACT: reduced to 200 words

Add a simple summary before the abstract, also simple summary must have a length of 200 words

https://www.mdpi.com/journal/animals/instructions

Be careful along the text to standardize the writing character

Move the paragraph Selenium characteristics in the soil and their influences on the aquatic environment at line 161

Rewrite references according instructions for authors

https://www.mdpi.com/journal/animals/instructions

Minor editing of English language required

Author Response

please find the attach cover letter

This manuscript is a resubmission of an earlier submission. The following is a list of the peer review reports and author responses from that submission.

Round 1

Reviewer 1 Report

The manuscript is a review on the effect of selenium on the growth and immune system of aquatic species. I find it useful for physiologists, zoologists, ecologists and geochemists.

I have a few comments about the manuscript:

1)      Title says " Effect of Selenium on the Growth and Immune Systems of Fish"

But the authors provide data on other biological effects associated with other processes. In particular, on the blood biochemistry not associated with the immune system

2)      Table 3: the center column title says "Se concentration (mg/kg)", but the table gives units for calculating the concentration. Either authors need to recalculate everything into single units, or change the heading of the table column.

3)      In the tables, the names of fish species are not uniform: some in the form of Latin names, some in the form of generally accepted names. I think it's better to do it in a single form.

4)      There are 138 items in the list of bibliography, but I saw only 100 in the article

Author Response

Response to reviewer 1

Reviewer 2 Report

This paper explored the effect of dietary selenium on the growth and immune systems of fish, which supply comprehensive sight on the dietary selenium effect on fish. However, there are some questions should be addressed.

1) In the abstract, of Line 13 “biological effects that include their growth performance and gene-related issues in many fish species”, “gene-related tissues” should be checked. Line70: “pathway mechanisms” can be removed. Others formats also should be checked according to the guild for the authors.

2) Line 86-94 maybe placed after Line123. Line180-189 maybe located before Line248.

3) Two graphs you can plot to improve the quality: one is about selenium effect on growth, immunology and so on. Another is about the excessive and deficiency effect on the aquatic species.

4) The difference between different species in response to environmental selenium can be discussed more.

5)the english language need to be improved.

Author Response

Response to reviewer 2

Reviewer 3 Report

Title: Effect of Dietary Selenium on the Growth and Immune Systems of Fish. Review

General comments

This paper reviews the positive and negative effects of selenium on fish and discusses potential sources of selenium in the aquatic environment. It explains that there is an optimal level of selenium necessary in the diet to support healthy biological processes of fish such as growth, reproduction, and immunity. It explains how extreme concentrations can be damaging to fish and higher levels of the food chain through bioaccumulation. I am unclear of the relevance of the article as it says it was written for the feed industry but I did not see any specific recommendations for producing feeds with selenium or how to administer them other than it is important to consider other potential sources of selenium ingestion in fish which may come from anthropogenic activities or other natural processes. It might be helpful to discuss how the feed industry could monitor selenium in their farmed fish and their aquatic environments. I am also not sure of the scope of the review. For instance, is selenium toxicity a common problem in farmed fish?

This paper overall is lacking in organization and clarity. There is information towards the end that would be good to know up front as the reader. The information in the paper is redundant in many places and does not flow well. There are inconsistencies in information presented (ie lines 229-230 vs 243-244), the methods for search and inclusion criteria of references are not included, and the main text is broad and lacking in focus. I do not believe it is specific or thorough enough for a review. It would be good for the reader if each reference was explained better in the text so the reader does not have to search and read the article being referenced. Tables are inconsistent. Terms are not always defined in the tables and there is inconsistent use of common/scientific names of fish species. I suggest a major revision of this review before publishing.   

Specific comments

Line 3: I believe the period in “Fish. Review” should be a colon?

Lines 12-13: “diverse sources and pathways getting into the aquatic environment” is redundant from first sentence. Suggest rephrasing or removing

Line 13: suggest changing “will cause” to “may contribute to”

Line 14: What kinds of gene-related issues? Upregulation? Downregulation? Both?

Line 14: Suggest using a different word than “diverse”. “Diverse” is used in line 11 and 12 right before this and there are a total of 17 usages of “diverse” in the entire manuscript. It feels overused and redundant.

Lines 17-18: Suggest changing “is detrimental and disrupts” to “can be detrimental and may disrupt”

Lines 18-19: The soil environment plays a greater role than what? This is not clear.

Lines 20-21: What were the methods for inclusion of references? What did you search and how did you select which articles/studies to include? I’m not sure this has to go in the abstract but it should be somewhere in the review paper.

Line 21: Suggest adding what types of populations (ie wild/hatchery/wild-farmed, etc.) after “development of the feed industry using dietary selenium…” so something like “development of the feed industry using dietary selenium for wild-farmed populations of fish…”

Line 22: Suggest adding how it promotes the conservation of the fish environment

Line 28: What kinds of “other processes”?

Line 28 and 29: Overuse of the word “very”. There are 12 uses in the article.

Line 29: “Fish are very selective for dietary selenium concentration…” I am unclear of what this means. I don’t think selective is the correct word choice here.

Lines 31-32: “They are very sensitive to the extreme introduction of selenium…” I am unsure where the extreme amounts of selenium are coming from. Is the feed industry putting too much selenium in their feeds? Are farmers overfeeding high concentrations of selenium when the fish are already being naturally exposed based on their environment?

Line 37: “the dietary selenium requirement for fish should be considered in fish feed formulation…” I’m not sure what the significance of this statement is. Is the amount of selenium in fish feeds not regulated?

Lines 38-39: “it is required in aquaculture and ranges between…” What is required in aquaculture? Selenium as a dietary requirement in feeds or regulation of selenium concentrations in feed?

Lines 40-43: I think this is the main point and should be moved up in the introduction. It is important for making it clear that feed should be regulated for selenium because fish in their natural environments may be exposed to excess selenium already depending on what they are exposed to.

Lines 50-54: Suggest more brevity as with a lot of sentences in this paper. Suggest shortening to something like “Selenium in the water may be taken orally or assimilated through the intestines and gills potentially causing bioaccumulation in fish muscles and organs.” Also suggest changing “organs” to be more specific. For instance, “in fish muscle and liver tissues”.

Line 55: “fish are the most sensitive” compared to what? Other trophic levels in the aquatic environment?

Lines 57-59: This sentence about effects of selenium does not go with this paragraph about modes and consequences of too much selenium intake ie bioaccumulation.

Line 63: Add scientific names of fish species

Lines 60-63: This sentence is confusing. “effects of selenium as a dietary requirement” implies the effects are going to be positive but then it goes on to state negative effects too ie bioaccumulation.

Lines 63-64: What causes dietary depletion of the hepatopancreas’ energy stores? Why is this relevant?

Lines 66-71: This is a great explanation of how selenium levels can be hazardous to fish. Suggest reorganizing the paper so all hazardous effects of selenium are discussed in the same section rather than in multiple places.

Lines 77-78: “which indirectly affects humans as end users and causes extremely toxic biological effects” The reference here does not explain toxicity of selenium to humans as far as I could tell. Is there a reference you could include for toxicity to humans and then also suggest adding what kinds of extremely toxic biological effects were observed.

Line 79: If this paper is focused on the “biological effects of extreme dietary selenium in fish” then I suggest only briefly mentioning the positive effects from selenium at optimal levels.

Line 87-88: “The environment plays a role in all metabolic processes necessary for fish survival.” Very broad, not sure the relevancy here.

Line 88: Suggest changing “man’s population” to “the human population”

Line 97: Suggest removing “(man)” and leaving “anthropogenic sources”

Lines 96-98: Information about anthropogenic sources is redundant from lines 40-43. Suggest talking about anthropogenic sources and other sources of selenium all together in one section.

Lines 105-106: This is the first mention of different forms of selenium. Suggest adding selenite and selenate with abbreviations (IV and VI) in line 94 when forms of selenium is first mentioned.

Line 108: Define EC50 since this is the first mention of the term.

Lines 108-109: Define difference between selenium IV and VI

Line 115: I think “implies” is the wrong word to use here. Suggest using a different word. 

Line 134: Fish species are toxic substances? Suggest rewriting this sentence for clarity.

Lines 152-155: This is a nice sentence highlighting the positive biological effects of selenium. Suggest talking about positive biological effects all together in one section instead of spread out in multiple sections.

Line 161: Suggest adding definition after “selenium” so changing to “selenium (Se)” so the term is defined already when used in the table.

Line 179: Define “Se” in each figure/table

Line 194: Center words “Aquatic Environment”

Lines 202-206: In Figure 1, suggest including reference for each effect.

Table 2: Define “Se” in description of table and suggest being consistent on use of common/scientific names

Lines 229-230: Change order to be lowest to highest or vice versa.

Line 233: Whose existence?

Lines 235-236: Suggest including scientific names

Line 240: Suggest using a different word than “depend”. I think this is the wrong use and it makes the sentence confusing. I think you are trying to say that soil type influences selenium concentrations?

Lines 243-244: This appears to be contradicting information for values of selenium in soils from what what stated in lines 229-230. Suggest combining the information and references into one sentence.

Lines 247-248: “selenium cannot just get into the aquatic environment by dietary means but also by runoff and other means” suggest rewriting or removing since sources of selenium into the aquatic environment has already been discussed. There are 5 mentions in the article of runoff being a source of selenium. Suggest summarizing this all in one place so it is less redundant.

Table 3: Define “Se” and check units for consistency

Lines 257-261: Redundant information

Line 266: Define “Se-Cys” as it is first time using

Line 268: “Adequate” and “deficiency” do not go well together. Suggest rewriting for clarity

Line 272: Define “GPx” as it is first time using

Line 285: Selenium plays a greater role than what?

Lines 286-287: “selenium is important for growth performance” This information is redundant. Already stated in lines 281-284 among other places.

Lines 287-290: Suggest adding common names. There is also inconsistency in italicizing scientific names.

Lines 295-306: This information would be good for the reader to know at the beginning of the paper.

Lines 333-334: Suggest using both common and scientific names for each species

Table 7: Define “Se” and suggest using both common and scientific names

Line 344: Suggest changing “man’s activities” to “human activities”

Line 365-366: Table 9 is referenced but there is no table 9

Line 377: What kinds of abnormalities and deformities?

Line 388: Suggest explaining what types of biological methods or techniques may be used

The English language is overall good, but there are times when a word is used in the wrong context and the point of the sentence gets misconstrued. 

Author Response

Response to specific comments 

Reviewer 4 Report

Dear authors

The topic of the paper is interesting. Selenium in fish can be seen on the one hand as a fundamental element in several biological processes, with defined levels. On the other hand, it can be toxic when at certain levels. This toxicity can be increased in certain locations, associated with anthropogenic activities and, in cases of aquaculture, entail additional care in feeding.

The theme of the paper is a strong point of the work. However, none of this is apparent in the paper as the ideas are very mixed and not easy for the reader to understand.

Perhaps start with a short introduction of the importance of selenium in general, in different animal species, including humans. Talk a bit more about the metabolic pathways affected, where glutathione peroxidase, thyroid, immune system, morbidity and mortality both due to deficiency and excess. 

Refer and reinforce the importance of fish in the study:

- as a model of toxicity

- indicators of the status of the aquatic environment and its importance in a global approach

- selenium deficiency in a global approach and relate it to a food source for humans

- one health approach of the theme

Although the topic is very interesting and current, the paper needs to be rewritten, organized and focused towards the goal.

Authors should not be discouraged as the idea is very good. It needs to be made appealing to the reader.

Here are some examples of sugestions

Abstrat

- the Title should be adapted - the article is on production pedixes or in their natural habitat

 - start with a one sentence introduction

- the first paragraph of the introduction has no references

- there should be a systematization of what is fish in their natural habitat and aquaculture production systems. It is confusing

- the introduction is rather disorganized. Why not put first the importance of selenium, then the sources and then supplementation or another more organized way?

- separate selenium needs from toxicity

-ex- line 60 - effects of selenium? Or effects of selenium deficiency or excess (since you talk about bioaccumulation. It is very confusing

- Define the purpose of the article. It is not clear

-

In the point "Different sources of selenium pathway mechanisms in the environment and their effects on fish" you start by talking about the oriegm of selenium in the environment. Then it talks about fish as a model for studying toxins and then again about selenium. It is very confusing. H+a have to organize by items in another way that is understandable.

- The different subjects are very mixed up.

- line 213: "The various forms, such as organic, 213 inorganic, and nanoforms, have an impact on how well fish species absorb dietary sele-214 nium" what does this sentence mean? 

- IN line item 211 - lots of confusion between environmental and dietary selenium? I don't understand what diet? In aquaculture?

Author Response

Response to reviewer 4

Reviewer 5 Report

Add a simple summary

Line 20: remove the bold of this

Carefully review the references along all  text because it is not possible to pass from reference number 22 in line 74 to reference number 28 in line 85 and become reference 103 in line 91

Line 27-78: this part is very confusing. I suggest to rewrite it in a more orderly way, first the importance of selenium, then the roles in fish metabolism, then the human operations issue and the bioaccumulation issues in fish

The chapter Different sources of selenium pathway mechanisms in the environment and their effects on fish is too confusing. Rewrite

Expand the chapter Forms of dietary selenium intake including the different possible administrations, eg granulated powder integrated with food

Expand the chapter Selenium distribution including in the known species the plasma levels of selenium and in which organs it is physiologically present, if possible also differences in sex

Extensive editing of English language required

Author Response

Response to reviewer 5
